# Single-Cell Analysis of ADSC Interactions with Fibroblasts and Endothelial Cells in Scleroderma Skin

**DOI:** 10.3390/cells12131784

**Published:** 2023-07-05

**Authors:** Marvin L. Frommer, Benjamin J. Langridge, Laura Awad, Sara Jasionowska, Christopher P. Denton, David J. Abraham, Jeries Abu-Hanna, Peter E. M. Butler

**Affiliations:** 1Charles Wolfson Centre for Reconstructive Surgery, Royal Free Hospital, London NW3 2QG, UK; b.langridge1@nhs.net (B.J.L.); jeries.abu-hanna@ndcls.ox.ac.uk (J.A.-H.); peter.butler1@nhs.net (P.E.M.B.); 2Department of Surgical Biotechnology, Division of Surgery & Interventional Science, University College London, London NW3 2QG, UK; 3Department of Plastic Surgery, Royal Free Hospital, London NW3 2QG, UK; 4Centre for Rheumatology, Department of Inflammation, Division of Medicine, University College London, London NW3 2QG, UK; 5Division of Medical Sciences, University of Oxford, Oxford OX3 9DU, UK

**Keywords:** single-cell RNA sequencing, scleroderma, skin fibrosis, autologous fat grafting, adipose-derived stem cells, ADSC, secretome, cellular therapies

## Abstract

Adipose-derived stem cells (ADSCs) as part of autologous fat grafting have anti-fibrotic and anti-inflammatory effects, but the exact mechanisms of action remain unknown. By simulating the interaction of ADSCs with fibroblasts and endothelial cells (EC) from scleroderma (SSc) skin in silico, we aim to unravel these mechanisms. Publicly available single-cell RNA sequencing data from the stromal vascular fraction of 3 lean patients and biopsies from the skin of 10 control and 12 patients with SSc were obtained from the GEO and analysed using R and Seurat. Differentially expressed genes were used to compare the fibroblast and EC transcriptome between controls and SSc. GO and KEGG functional enrichment was performed. Ligand–receptor interactions of ADSCs with fibroblasts and ECs were explored with LIANA. Pro-inflammatory and extracellular matrix (ECM) interacting fibroblasts were identified in SSc. Arterial, capillary, venous and lymphatic ECs showed a pro-fibrotic and pro-inflammatory transcriptome. Most interactions with both cell types were based on ECM proteins. Differential interactions identified included *NTN1*, *VEGFD*, *MMP2*, *FGF2,* and *FNDC5*. The ADSC secretome may disrupt vascular and perivascular inflammation hubs in scleroderma by promoting angiogenesis and especially lymphangiogenesis. Key phenomena observed after fat grafting remain unexplained, including modulation of fibroblast behaviour.

## 1. Introduction

Scleroderma (systemic sclerosis, SSc) is a complex autoimmune connective tissue disease characterised by extensive and progressive skin and internal organ fibrosis due to excessive collagen deposition, microvascular damage, and a dysfunctional immune response [1,2]. As a paradigm of fibrotic skin diseases, it is characterised by inflammation driving the excessive accumulation of extracellular matrix (ECM) components in the skin, which leads to thickening and stiffening, decreased skin elasticity, loss of function, and an altered appearance. The initial and significant manifestations of SSc usually involve vascular malfunctions and abnormalities. The microvascular dysfunction can lead to discolouration, skin ulceration, and Raynaud’s phenomenon [1,2]. Structures such as sweat glands and hair are lost, as well as the underlying adipose tissue, due to chronic inflammation [3]. In scleroderma, hardening and thickening of the skin especially affect the face and extremities [1,2]. Symptoms such as microstomia significantly reduce the quality of life [4]. Patients report that the impact of the disease on face and hands ranked higher than cardiac, renal, and lung, and were the most impacting on their quality of life [4].

Fibroblasts play a critical role in scleroderma and the pathophysiology of other fibrotic skin diseases [5]. Dysregulated fibroblast activation leads to the excessive secretion of ECM components such as collagen and fibronectin. Targeting these cells was proposed as a therapeutic strategy for treating fibrotic diseases. The advances in next-generation sequencing granted new means of exploring complex biological systems [6]. Single-cell RNA sequencing (scRNA-seq) enables the discovery of disease-driving cell subpopulations, tracking the trajectory of cell lineage as well as analysing underlying pathways and regulatory relationships [6]. This led to the discovery of functionally and anatomically distinct fibroblast subpopulations in normal skin [7]. In scleroderma, dermal fibroblast populations were compared between SSc and healthy skin, demonstrating transcriptomic changes and how SSc dermal myofibroblasts could arise from a specific progenitor fibroblast population [8]. Naturally, much effort was recently invested into exploring disease-driving fibroblast subpopulations in various chronic inflammatory and fibrotic diseases, as this could possibly lead to the discovery of targeted therapies. A scRNA-seq study recently provided evidence that there may be a shared pro-inflammatory fibroblast phenotype across conditions rather than tissue and disease-specific fibroblasts [9,10].

Vascular involvement may be a crucial and fundamental process in the development of scleroderma, which is strongly supported by clinical and pathologic observations of early vascular damage and endothelial activation [11]. The vascular alterations observed in SSc are more in line with vasculopathy than vasculitic processes, primarily due to the limited presence of inflammation in the vessel wall but extensive systemic intimal proliferation with luminal obstruction and lymphocyte infiltration [12]. Vessels exhibit signs of vascular dysfunction, including impaired tone, vascular permeability, and vasospasms. Further, although plasma levels of pro-angiogenic growth factors are elevated, angiogenesis is defective and marked by a loss of capillaries and small vessels [13]. Single-cell analysis of endothelial cells in scleroderma demonstrated upregulation in pathways associated with extracellular matrix generation and negative regulation of angiogenesis [14].

Immunosuppressive strategies, including autologous haematopoietic stem cell transplantation, appear beneficial in treating scleroderma, but a high unmet need remains, especially for therapies that can attenuate fibrosis. However, autologous fat grafting (AFG) shows great promise in treating a variety of fibrotic skin conditions, including scleroderma [15,16,17,18]. It is a commonly used reconstructive technique for correcting volumetric tissue deformities and histologically led to the reduction, remodelling, and realignment of collagen fibres, increased vascularisation, and reduction in α-SMA, dermal thickness, and fibrotic defect size [19]. It also enhanced aesthetic appearance with less discolouration and reduced painfulness [20,21]. The exact mechanism by which AFG improves fibrosis remains unknown. Fat tissue mainly consists of adipocytes, but a variety of other cells are also present that are collectively named stromal vascular cells. The stromal vascular fraction (SVF) consists of a mix of cell types, including a variety of immune cells, endothelial cells, smooth muscle cells, and adipose-derived stem cells (ADSCs) [22]. ADSCs are stem cells of mesenchymal origin and are believed to be of central importance in the AFG’s anti-fibrotic and anti-inflammatory effects [19,21]. Their potential as a treatment for skin fibrosis is due to their ability to differentiate into multiple cell types and modulate the immune system [23,24]. Further, ADSCs were shown to reduce ECM accumulation, inhibit fibroblast activation, and promote revascularisation. Their paracrine signalling is hypothesised to be the paramount mechanism by which they counteract inflammation and skin fibrosis and promote tissue repair. For example, indirect co-culture of ADSCs with fibroblasts from scleroderma led to a decrease in the secretion of transforming growth factor (TGF-β_1_) and connective tissue growth factor (CTGF) [16].

In this study, we explore scRNA-seq data to gain further insight into the biological niche of ADSCs and their transcriptome and functionalities. Subsequently, we analyse the interactions between ADSCs and disease-driving fibroblast and endothelial cell subpopulations from the skin of patients with scleroderma. These interactions could underpin the therapeutic effects of fat grafting in fibrotic skin diseases and further investigation may lead to the development of new targeted treatments including pharmaceutical and cell-based therapies.

## 2. Methods

### 2.1. Data Acquisition and Patient Characteristics

This study utilised publicly available RNA sequencing datasets. Single-cell data of white adipose tissue (WAT) were taken from the gene expression omnibus (GEO) with the accession GSE155960. The datasets of three lean patients (BMI < 25) without systemic disease were accessed. All three lean donors were female with an average BMI of 20.73 and an average age of 38 years. The patient samples were processed to obtain the SVF and purified using fluorescence-activated cell sorting (FACS). Additional information on library preparation, sequencing, and alignment of the single-cell data can be found in Hildreth, et al., 2021 [25]. Single-cell data from the skin of twelve patients affected by scleroderma (SSc) and ten healthy control subjects were taken from the GEO with the accession GSE138669. All samples were taken from dorsal mid-forearm biopsies. All scleroderma patients had diffuse cutaneous disease with a variable disease duration of 0.48 to 6.48 years (mean = 2.42 years) and a modified Rodnan skin score (MRSS) of 12 to 43 (mean = 26.17). Additionally, gender (control = 5/10 female; SSc = 7/12 female) and age (control mean age 51.9; SSc mean age = 54.7) were reported. Additional information on data procurement can be found in Tabib et al., 2021 [8].

### 2.2. Pre-Processing and Preparation

The R package Seurat (v.4.3.0) was used for analysis [26]. After creating Seurat objects from the raw count data matrices of the individual samples, cells were assessed for their quality. Cells that expressed > 25% mitochondrial genes and had <500 gene transcripts were filtered out as low-quality cells. Individual samples were processed with SCTransform (v.0.3.5) using the glmGamPoi method (v.1.10.2) [27,28]. SCTransform removes the influence of technical characteristics from downstream analyses while preserving biological heterogeneity. Further, it omits the need for heuristic steps such as log transformation or pseudocount addition. Individual samples of each dataset were integrated based on the SCT assay slot [29], and 3000 features were selected to perform the integration.

### 2.3. Cell Clustering and Cell Type Annotation

After pre-processing and preparation, principal component analysis (PCA) was performed on the variable genes. A total of 30 PCs were used for subsequent clustering and uniform manifold approximation and projection (UMAP) dimensional reduction. To identify clusters of cells, a shared nearest neighbour (SNN) modularity optimisation-based clustering algorithm was applied [30]. The resolution to acquire the cell clusters varied between datasets (WAT = 1.5; skin = 0.7). FindAllMarkers was used to calculate differently expressed genes (DEGs) between clusters, and the resulting cluster markers were used to manually annotate cell types. Manual cell type annotation was performed with the human protein atlas and previous single-cell and transcriptomics data [7,31,32,33,34,35,36,37]. In the WAT dataset, sub-clustering was performed with the FindSubCluster function (res = 1.0) to divide the lymphoid cell subpopulations further. In the skin dataset, sub-clustering was performed on fibroblasts (res = 0.2) and endothelial cells to first identify lymphatic endothelial cells (res = 0.2) and then subsequently sub-cluster blood endothelial cells (res = 0.3). Correlation matrices based on Pearson’s product-moment correlation coefficient (r) exploring the overlap of highly variable genes between the clusters were used to confirm the sub-grouping. For calculating the proportions of each cell type, the number of cells per cluster was divided by the total number of cells per sample.

### 2.4. Cellular State Plot

We explored the cellular states of the structural cells in the SVF by using Seurat’s function AddModuleScore, which provides enrichment scores for a manually defined list of genes [38]. Adipogenic differentiation markers chosen were apolipoprotein D (*APOD*), insulin-like growth factor 1 (*IGF1*), and chemokine ligand 14 (*CXCL14*) [32,39]. Vasculogenic differentiation markers chosen were CD105 (*ENG*) and CD31 (*PECAM1*) [32]. Myogenic differentiation markers chosen were smooth muscle alpha (α)-2 actin (*ACTA2*), regulator of G protein signalling 5 (*RGS5*), and transgelin (*TAGLN*) [32].

### 2.5. Gene Set Enrichment Analysis

Gene ontology (GO) and Kyoto Encyclopedia of Genes and Genomes (KEGG) functional enrichment (over-representation) was performed with the R package clusterProfiler [40,41,42,43]. The top 100 DEGs at an adjusted p-value lower than 0.05 were used as input for each cluster.

### 2.6. Cell-Cell Ligand–Receptor Interaction Analysis

LIANA (v.0.1.8) was used to explore cell–cell ligand–receptor interactions between ADSCs from WAT and fibroblast and endothelial cell subpopulations from scleroderma [44]. LIANA enables the analysis and integration of the results across multiple tools for analysing cell interactions and provides a meta-analysis of the co-expression of ligand–receptor pairs. Here, we used the Network Analysis Toolkit for Multicellular Interactions (NATMI; Hou, et al., 2020), Connectome (Raredon, et al., 2022), iTALK (Wang, et al., 2019), SingleCellSignalR (Cabello-Aguilar, et al., 2022), and CellphoneDB (Efremova, et al., 2020) to analyse cell–cell ligand–receptor interactions of ADSCs with fibroblasts and endothelial cells. We performed a unilateral analysis with ADSCs expressing the ligand and fibroblasts and endothelial cells expressing the receptor. Results are ordered by aggregate rank and serve as a consensus across a selected method based on the R package RobustRankAggreg. Further, the specificity of interactions as defined by NATMI’s edge specificity weights and expression magnitude based on SingleCellExperiment’s LRScore are reported. Specificity ranges from 0 to 1, where 1 means both the ligand and receptor are uniquely expressed in each pairing of cell types, while expression magnitude represents a regularised score comparable between datasets.

### 2.7. Data Visualisation

Data visualisation of single cell results was performed using R (v.4.2.2) and Seurat and the packages SCpubr (v.1.1.2), ggplot2 (v.3.4.0), cowplot (v.1.1.1), and pheatmap (v.1.0.12) [45,46].

### 2.8. Statistical Analysis

For comparing proportions of cell types, statistical analysis was performed with GraphPad Prism 9 (GraphPad Software, San Diego, CA, USA). Data in the results are reported as means ± standard deviation and *n* represents the number of patients. A Mann–Whitney U test was used to evaluate statistical differences.

## 3. Results

### 3.1. The Diverse Single-Cell Landscape of the Stromal Vascular Fraction

The quality-controlled single-cell atlas of the stromal vascular fraction (SVF) of white adipose tissue (WAT) from three lean patients was composed of 38,391 cells. Analysis of differential expression of marker genes and visualisation with a uniform manifold approximation and projection (UMAP) plot revealed 18 distinct clusters: adipose-derived stem cells (ADSC), adipocyte precursor cells (APC), endothelial cells (EC), smooth muscle cells (SMC), B cells (BC), regulatory T cells (TREG), mucosal-associated invariant T cells (MAIT), naïve CD4 T cells (nCD4 T), naïve CD8 T cells (nCD8 T), mature CD8 T cells (mCD8 T), natural killer-like cells (NK-like), mature natural killer cells (mNK), innate lymphoid cells (ILC), lymphoid-like cells (lymphoid-like), classical dendritic cells (cDC), non-classic monocytes (ncMos), myeloid-like cells (myeloid-like), and macrophages (Mac) (Figure 1A). The clusters labelled lymphoid-like and myeloid-like cells could not be defined further. Marker genes corroborated from previous single-cell analyses [7,31,32,33,34,35,36,37] and DEG analysis were used for cell type annotation (Figure 1B,C and Appendix A). The missing expression of *Leptin* (*LEP*) indicates the absence of adipocytes in the dataset. The SVF consists of a variety of cell types, including a diverse system of immune cell lineages. This strengthens the assumption of adipose tissue being an immunologically active organ and its ability to drive systemic inflammation in obesity [47]. Some notable variability of cell type proportions exists between the three donors (Figure 1D).

### 3.2. The Transcriptomic Profile and Functions of Adipose-Derived Stem Cells

Of the SVF 1285 cells were identified as ADSCs (Figure 2A). The IFATS recommends *CD73*, *CD90*, *CD34*, *CD44*, *CD29,* and *CD105* as cell surface markers to identify and characterise ADSCs in cell culture [39]. Further, a systematic review suggested *CD59* as positive and *CD56*, *CD62*, and *CD45* as negative markers [48]. Expression of *CD73*, *CD90*, and *CD34* in ADSCs was high compared to the other structural cells (APC, EC, and SMC) (Figure 2B). There was little to no expression of *CD105*; notably, here we explored ADSCs naïve to cell culture. Additionally, among the structural cells of the SVF, *CD26* showed specific expression in ADSCs. When computing a cell state plot based on adipogenic, vasculogenic, and myogenic differentiation, ADSCs were located in the middle, indicating no commitment to either of these lineages (Figure 2C).

To gain more insight into the diverse functions of ADSCs, enrichment for GO and KEGG pathways was performed (Figure 2D). ADSCs were especially enriched for pathways associated with the ECM, including extracellular matrix binding, fibronectin binding, and ECM receptor interactions. These cells further showed enrichment for antioxidant activity and cytokine binding. We wanted to investigate specific genes associated with anti-fibrotic and anti-inflammatory effects of ADSCs from the literature. Genes were selected from the literature and their expression was plotted (Figure 2E) [19,20,21]. For example, we selected hepatocyte growth factor (*HGF*), which downregulates transforming growth factor-β and promotes angiogenesis [49]. However, expression of *HGF* at the single-cell level could not be confirmed. ADSCs expressed genes associated with ECM functions, including cellular communication/matricellular factors (CCNs), collagen subtypes, MMPs, TIMPs, and members of the disintegrin and metalloproteinase with thrombospondin motif (ADAMTS) family. Single-cell ADSCs were further enriched for a variety of angiogenic factors, including *VEGFB*, *VEGFD*, *ANG*, *ANGPTL2*, and *ANGPTL5* (Figure 2E).

### 3.3. The Paradigm of Fibrotic Skin Diseases—Scleroderma

After quality control, we obtained 38,391 cells from the skin of ten control and twelve scleroderma patients. Clustering revealed 13 clusters: fibroblasts (FB), basal keratinocytes (BKC), differentiating keratinocytes (DKC), melanocytes (MEL), neural cells (NC), glandular cells (GL), endothelial cells (EC), smooth muscle cells (SMC), pericytes (PC), T cells (TC), dendritic cells (DC), macrophages (Mac), and mast cells (MC) (Figure 3A). Marker genes were corroborated from the human protein atlas and previous single-cell analyses [7,31,32,33,34,35,36,37] in combination with DEG analysis and were used for cell type annotation (Figure 3B,C and Appendix A). Proportionally, there is a relative depletion of fibroblasts, endothelial cells, and pericytes in scleroderma samples compared to control, while a relative increase in keratinocytes can be observed (Figure 3D). However, the differences do not reach significance following statistical analysis (Appendix A).

### 3.4. Pro-Inflammatory and ECM-Interacting Fibroblasts in Scleroderma

Sub-clustering of 9192 cells identified as fibroblasts revealed six functionally distinct clusters (Figure 4A). No significant changes in fibroblast subcluster proportions were found between control and scleroderma; however, cluster FB5 was missing from both control and scleroderma patients in some cases (Figure 4B and Appendix A). There is no universal classification of fibroblast subtypes, and hence no generally accepted method for the identification of subgroups at the single-cell level exists. Neither approach to classify them anatomically nor functionally is generally accepted. However, Ascensión, et al. [7] recently conducted a pooled analysis of various single-cell datasets of human dermal fibroblasts to identify commonly shared clusters. Three major groups were identified: dermal cell and ECM homeostasis (A), immune surveillance and pro-inflammatory (B), and a variety of specialised subpopulations (C). After obtaining our subgroups from Seurat’s self-guided clustering algorithm, we found that both control and scleroderma fibroblast subclusters fit this classification system (Figure 4C). Clusters FB3 and FB4 were identified as dermal cell and ECM homeostasis (A) subtypes, cluster FB1 was identified as an immune surveillance and pro-inflammatory subtype (B), and clusters FB2 and FB5 as specialised subpopulations (C). Specialised subpopulations can be fibroblasts enriched in aged skin or located at the dermal papillae or the dermo-hypodermal junction [7]. FB6 less clearly matched the classification but showed the closest resemblance to the immune interacting group. Results of DEG analysis and minimal overlap of highly variable genes between the clusters support the subgrouping results (Appendix A).

Subsequently, we compared the transcriptomic profiles of the subclusters between each other and between control and scleroderma (Figure 4D). Certain genes were highly expressed in scleroderma while being absent in control fibroblasts and vice versa. FB1 expresses high levels of chemokine (C-C motif) ligand 19 (*CCL19*), phospholipase A2 (*PLA2G2A*), and metallothionein 2A (*MT2A*). CCL19 plays a crucial role in regulating the induction of T-cell activation, immune tolerance, and inflammatory responses [50], while PLA2G2A is associated with inflammation and both lung and cardiac fibrosis [51,52]. Adding to this, FB4 in scleroderma expressed higher levels of *THY1* and fibroblast activation protein-α (*FAP*) (Figure 4D). FAP + THY+ fibroblasts were found to be responsible for severe and persistent inflammation in arthritis [10]. In accordance with the disease hallmarks, subtypes of collagen (*COL1A1*, *COL6A1*) were more highly expressed in scleroderma. Additionally, a variety of other genes associated with inflammatory as well as malignant diseases were enriched in scleroderma fibroblasts (*POSTN*, *PRSS23*, *TNC*, and *SERPINE2*). Analysis of enriched GO and KEGG pathways supports the functional classification in immune and ECM subtypes (Figure 4E,F). Scleroderma fibroblasts display higher enrichment scores for pathways regarding chemokine, cytokine, and immune receptor activity. FB6_SSc_ showed no enrichment for KEGG pathways (Figure 4F).

### 3.5. Endothelial Cells in Scleroderma Promote Inflammation

Sub-clustering of 3282 cells identified as endothelial cells revealed four distinct clusters (Figure 5A). Arterial endothelial cells (AEC), capillary endothelial cells (CEC), venous endothelial cells (VEC), and lymphatic endothelial cells (LEC) were identified. Proportions did not differ significantly between control and scleroderma (Figure 5C and Appendix A). He, et al. [37] analysed the transcriptomic profile of dermal blood endothelial cells to enable the identification of subclusters. We adapted their approach to our dataset and used their marker array for the annotation of our clusters (Figure 5C). Additionally, LECs were identified by expression of lymphatic vessel endothelial hyaluronan receptor 1 (*LYVE1*) and podoplanin (*PDPN*) (Figure 5C).

Then, we compared the transcriptomic profiles of the subclusters between each other and between control and scleroderma (Figure 4D). Von Willebrand factor (*VWF*) and Von Willebrand factor A domain containing 1 (*VWA1*) are responsible for platelet adhesion and clot formation and were more highly expressed in scleroderma compared to control, indicating possible causes for microvascular damage. Genes associated with ECM components, such as *COL4A1*, heparan sulfate proteoglycan 2 (*HSPG2*), and actin beta (*ACTB*), as well as phospholipase C gamma 2 (*PLCG2*), fatty binding protein 4 (*FABP4*), and galectin 1 (*LGALS1*), also showed higher expression levels in scleroderma. Control endothelial cells express higher levels of cytokines involved in immune cell recruitment and inflammation, such as C-C motif chemokine ligand 2 (*CCL2*), interleukin 6 (*IL6*), and colony stimulating factor 3 (*CSF3*). Analysis of GO pathways showed similar enrichment, but endothelial cells from scleroderma showed higher enrichment scores for chemokine activity, platelet-derived growth factor binding, and integrin binding (Figure 5E,F). While in control, endothelial cell KEGG pathways were primarily enriched in CECs. In scleroderma, AECs showed the most enriched KEGG pathways (Figure 5E,F). The IL17 signalling pathway was downregulated in scleroderma, while leukocyte transendothelial migration, vascular smooth muscle contraction, and the pro-inflammatory PPAR signalling pathway were upregulated. LECs showed no enrichment for KEGG pathways in either control or scleroderma (Figure 5E,F).

### 3.6. Cellular Communication Networks after in silico ADSC Treatment

There are established cellular and molecular differences between healthy and scleroderma skin. To understand how ADSCs impact and possibly treat fibrotic skin in scleroderma patients, we analysed cellular communication networks through ligand–receptor analysis using LIANA [44]. The interactions between ADSCs and cells from healthy skin served as a control to understand physiologic and homeostatic communication and identify differences in interactions with cells from scleroderma.

Between subclusters of healthy and scleroderma skin, 49.29% (35/71) of the interactions were shared. Most of the top interactions between ADSCs and fibroblast subtypes in both control and scleroderma are based on cellular communication network factors (CCN) and integrins (Figure 6 and Appendix A). Integrin subunit alpha V (*ITGAV*) and caveolin 1 (*CAV1*) interactions were specific to scleroderma fibroblasts (Figure 6C). Similarly, thrombospondin 1 (*THBS1*) activates latent TGF_β,_ increasing collagen production, and was exclusively present among the top interaction in the small FB6_SSc_ cluster [53]. The TNFSF9–HLA-DPA1 interaction is unique to the immune-interacting FB1_SSc_. Basic fibroblast growth factor (FGF2) interacted with FB2_SSc_ and FB5_Both_. Parathyroid hormone-like hormone (*PTHLH*) was expressed by ADSCs and interacted with almost all fibroblast clusters via receptor activity-modifying protein 2 (*RAMP2*). C-X-C motif chemokine ligand 12 or stromal-derived factor 1 (*CXCL12*) was only among the top interactions with FB4_SSc_. ADSCs also expressed fibronectin type III domain-containing protein 5 (*FNDC5*), which interacted with ITGB5 and additionally with ITGAV in scleroderma on various subclusters.

Most interactions between endothelial cells from both scleroderma and control skin are based on interactions of extracellular matrix proteins expressed by ADSCs including different collagen subtypes and proteoglycans (Figure 7 and Appendix A). More interactions with VECs, CECs, and LECs from scleroderma are based on the homing cell adhesion molecule (*H-CAM*, *CD44*) expressed by endothelial cells (Figure 7), which is among the various adhesion molecules upregulated in the serum of SSc patients [54]. In control skin, only AECs interacted through *CD44*. Matrix metalloproteinase-2 (*MMP2*), highly expressed in ADSCs (Figure 2E), interacts with endothelial cells from both datasets. Further, ADSCs interact with all EC subtypes except for AECs in both control and SSc via vascular endothelial growth factor D (*VEGFD*) to neuropilin 2 (*NRP2*). VEGF-D promotes both angiogenesis and lymphangiogenesis [55]. Interactions of retinoic acid receptor responder protein 1 (*RARRES1*) and *VEGFA* with *NRP2* were only among the top interactions in control ECs. ADSCs possibly interact with the beta-2 adrenergic receptor (ADRB2) on AECs and CECs from SSc to *PTHLH*. The interaction of tumour necrosis factor superfamily member 4 (TNFSF4) with its receptor TNFRSF4 is unique to the AECs from SSc. TNFSF4 is overexpressed in the skin and serum of patients with SSc, particularly in patients with diffuse cutaneous forms [56]. LECs from SSc are the only ECs to strongly interact with ADSCs through vascular endothelial growth factor receptor 3 (*FLT4*).

## 4. Discussion

Adipose-derived stem cells (ADSCs) and their secreted factors are believed to play a pivotal role in the observed anti-fibrotic and immunomodulatory effects of autologous fat grafting for treating fibrotic skin in scleroderma patients [16,57]. ADSCs are mesenchymal stem cells that can be isolated from fat tissue by manual and enzymatic digestion. Their paracrine signalling is believed to be the main mechanism of action by which they counteract inflammation and fibrosis, rather than their direct cell-to-cell contact interactions or regenerative, multilineage potential [19]. In vitro models with indirect co-culture of ADSCs or the use of ADSC-conditioned media on cells from fibrotic tissue demonstrated anti-inflammatory effects, reduced production of ECM components, inhibition of fibroblast activation, and promotion of revascularisation [23,24]. Analysis of the ADSC transcriptomic profile revealed enrichment, especially for genes associated with angiogenesis, as well as pathways associated with remodelling of the ECM (Figure 2). For example, VEGFD and VEFGB were almost exclusively expressed in ADSCs.

In our analysis of fibroblasts, FB1 was characterised as a pro-inflammatory or immune-interacting subtype by *APOE*, *CYGB*, *C7*, and *IFGBP7* and this subset showed higher levels of *CCL19* and *PLA2G2A* in scleroderma compared to the control (Figure 4) [7]. CCL19 along with CXCL10 positivity was also a distinctive marker of inflammatory fibroblasts shared across various chronic inflammatory diseases investigated by Korsunsky, et al., including rheumatoid arthritis, ulcerative colitis, interstitial lung disease, and Sjögren’s syndrome and afterwards confirmed in atopic dermatitis [9]. However, *CXCL10* was not highly expressed in FB1 (Appendix A). CCL19 plays a crucial role in the immune system’s response to inflammation and infection. CCL19 acts as a chemoattractant for immune cells such as T cells and dendritic cells, guiding them to sites of inflammation, and plays a crucial role in regulating the induction of T-cell activation, immune tolerance, and inflammatory responses [50]. In scleroderma, expression of *CCL19* correlates with vascular inflammation of the skin [58]. Korsunsky, et al. [9], further found a SPARC^+^COL3A1^+^ vascular-interacting fibroblast, which primarily corresponds to our ECM-interacting clusters FB3 and FB4 (Figure 4 and Appendix A). SPARC^+^COL3A1^+^ fibroblasts colocalised with arterial blood vessels and may play a role in vascular remodelling in inflammation [9]. Their expansion preceded the growth of CCL19^+^ immune-interacting fibroblasts, suggesting a two-stage mechanism for fibroblast-mediated regulation of inflammation, initiated by vascular remodelling that enables greater leukocyte infiltration into the tissue [9]. Tabib, et al. [8,59] also found that the fibroblast cluster expressing high levels of *CCL19* localised near the vasculature, as opposed to the control cluster, with low expression of *CCL19*. This serves as further evidence that fibrosis in scleroderma develops in close proximity to the vasculature [60]. It was also proposed that perivascular mesenchymal cells may serve as precursors to profibrotic myofibroblasts [61,62,63]. Further, FB4 in scleroderma expressed higher levels of *THY1* and *FAP* (Figure 4D). FAP^+^THY^+^ fibroblasts were involved in severe and persistent inflammation in arthritis [10]. We demonstrated that both inflammation-driving subtypes, CCL19^+^ and SPARC^+^COL3A1^+^/FAP^+^THY^+^, are present in scleroderma, possibly promoting inflammation through chemotaxis and enhanced leucocyte infiltration. Our discovery provides additional evidence that inflammation-associated fibroblasts might share activation states across chronic inflammatory diseases, leading to a common phenotype rather than being disease and tissue-specific. While the underlying activating factors could vary depending on the disease and tissue, two fibroblast states might be universal to inflammatory diseases across different tissues, as suggested by Korsunsky, et al. [9]. This could lead to targeted therapies across different inflammatory and fibrotic diseases.

The interactome between ADSCs and fibroblast subtypes in both control and scleroderma shows top interactions mainly based on CCNs and integrins (Figure 6). In general, CCN1 and CCN2 are believed to exert pro-fibrotic effects [64]. These effects depend on the engaged integrin subunit. CCN1 promotes cell proliferation, survival, and angiogenesis through ITGAV, while CAV1 regulates CCN1 secretion [65]. *ITGAV* and *CAV1* interactions were only among the top interactions in scleroderma fibroblasts (Figure 6C). Other ligands, such as tenascin XB (*TNXB*), *PTHLH*, and *CXCL12,* were shown to activate latent TGF-β, cause epithelial-to-mesenchymal transition (EMT), and increase fibroblast survival [66,67,68,69]. ADSCs also expressed *FNDC5*, a fibronectin, which interacted with ITGB5 and additionally with ITGAV in scleroderma. FNDC5 is associated with anti-inflammatory and anti-oxidative effects [70]. The TNFSF9–HLA-DPA1 interaction is unique to the immune-interacting FB1_SSc_; however, little is known about its effect. TNFSF9 is involved in the antigen presentation process and generation of cytotoxic T cells, possibly leading to increased inflammation [71]. Basic fibroblast growth factor (*FGF2*) is one of the major mechanisms described in the literature by which ADSCs modulate fibroblast activity [19]. Studies suggest it inhibits fibroblast-to-myofibroblast transition, causes myofibroblast apoptosis, and hence decreases the production of collagen by fibroblasts [19]. We found signalling from ADSCs via *FGF2* to *SDC1* on FB5 in both datasets and FB2 in scleroderma. However, neither of these subclusters are involved in inflammation nor primarily responsible for ECM production. Further, although SDC1 is a transmembrane heparan sulphate proteoglycan and possesses affinity to FGF2, the downstream effect of this interaction for fibroblast behaviour is unclear. Usually, FGF2 exerts its effects by binding to fibroblast growth factor receptors (FGFRs) [72]. It is possible that FGF2 does interact with other fibroblast subclusters, but those interactions were not highly specific or significant in our analysis. FG2 and HGF are believed to be part of an antifibrotic bFGF-JNK-HGF pathway [73]. HGF is proposed to be responsible for a variety of anti-fibrotic effects of ADSCs [19]; however, according to our scRNA-seq data, HGF is not strongly expressed by any cell type in the SVF (Figure 2). This suggests that either ADSCs start to express and produce HGF during in vitro culture, but not in their natural tissue environment, or they only start producing it in response to fibroblasts during co-culture. Notably, our study findings align with previous research that demonstrates significant alterations in the MSC phenotype in vitro, with a gain of *CD105* and loss of *CD34* expression (Figure 2B), which may explain the missing expression of effector molecules from the literature, such as *HGF* [74]. *CD34* expression is highly dependent on cell culture conditions, with decreased expression after prolonged in vitro cell culture [39]. ADSCs in the scRNA are naïve to cell culture, which could explain the difference when compared to commonly observed markers of cultured ADSCs [48]. It is also possible that in vitro cultures from the literature contained a mixed cell population, and other cell types besides ADSCs may be responsible for the measured HGF expression. In summary, ADSCs interact distinctly with fibroblasts from scleroderma. While certain interactions are specific to subsets, the possible derived effect remains unclear. FNDC5 and FGF2 could be responsible for observed effects from in vitro and in vivo studies; however, FGF2 was only found amongst the top interactions with FB2 and FB5, which are neither proinflammatory nor ECM-interacting. Most significant interactions are based on ECM components such as CCNs, collagens, tenascins, and fibronectin. Rather than directly acting upon fibroblasts via these molecules, ADSCs might change the composition of the ECM and reinstate tissue homeostasis [21,75]. Consequently, the altered ECM can influence the behaviour of fibroblasts through molecular and mechanotransduction processes.

While studies on fibroblast heterogeneity increasingly focus on functional states rather than anatomical location, it is logical to classify endothelial cells according to the vessel. We used the markers proposed by He, et al. [37]; however, we aimed for a broader anatomical classification to not overcomplicate the analysis and therefore divided ECs into arterial, capillary, venous, and lymphatic. Endothelial dysfunction is a hallmark feature of scleroderma and is characterised by reduced vasodilation, increased vasoconstriction, and impaired angiogenesis. Endothelial damage can lead to increased expression of adhesion molecules, which promote the migration and activation of immune cells into the tissue, further escalating the inflammatory response [11,12]. Further, although plasma levels of pro-angiogenic growth factors are elevated, angiogenesis is defective and marked by a loss of capillaries and small vessels [13]. As previously stated, fibrosis in scleroderma develops near the vasculature, and perivascular mesenchymal cells may serve as precursors to profibrotic myofibroblasts [60,61,62]. A single-cell analysis of endothelial cells in scleroderma found upregulation in pathways associated with ECM generation, negative regulation of angiogenesis, and epithelial-to-mesenchymal transition (EMT). Perlecan (*HSPG2)* was elevated in scleroderma ECs compared to the controls and the perivascular regions stained robustly for HSPG2 [14]. HSPG2 is a main component of the blood vessel basement membrane and is implicated in a variety of fibrotic diseases, including liver fibrosis [76]. It was found to be a fibrogenic mediator produced by SSc endothelial cells undergoing apoptosis and induced resistance to apoptosis in fibroblasts, and was found to cause myofibroblast differentiation. Apoptosis of endothelial cells is considered a key disease-driving mechanism in scleroderma [77]. While Apostolidis, et al. [14] compared endothelial cells as a whole, we found *HSPG2* to be upregulated especially in AECs and VECs (Figure 5). *PLCG2*, *FABP4*, and *LGALS1*, which are involved in inflammatory pathways as well as pathways regarding leukocyte transendothelial migration, vascular smooth muscle contraction, and the pro-inflammatory PPAR signalling pathways that were upregulated were also upregulated in scleroderma ECs (Figure 5) [78,79,80]. These findings underline the importance of vascular dysfunction and the vascular and perivascular niche for the pathogenesis of scleroderma.

Similarly to what we observed for fibroblasts, most interactions between endothelial cells from both scleroderma and control skin are based on interactions of extracellular matrix proteins expressed by ADSCs including different collagen subtypes and proteoglycans (Figure 7). ECM proteins expressed by ADSCs interacted with homing cell adhesion molecules (H-CAM, CD44) on VECs, CECs, and LECs from scleroderma (Figure 7). In control skin, only AECs expressed CD44, which is among the various adhesion molecules upregulated in the serum of SSc patients [54]. Through interactions with hyaluronic acid and other ligands, CD44 promotes adhesion, migration, proliferation, and survival of endothelial cells, resulting in increased angiogenesis [81]. Matrix metalloproteinase-2 (MMP-2), highly expressed in ADSCs (Figure 2E), interacts with endothelial cells from both datasets. In general, MMPs can promote EC migration and tube formation [82]. Further, ADSCs interact with all EC subtypes except for AECs in both control and SSc via vascular endothelial growth factor D (VEGFD). VEGF-D promotes both angiogenesis and lymphangiogenesis [55]. ADSCs further interacted with LECs via NTN1 to UNC5B. Netrin 1 (*NTN1*) is part of the laminin family and is involved in axonal growth and direction during embryogenesis. It was further found to promote angiogenesis by blocking apoptosis induced by UNC5B—the apoptosis of ECs being a key mechanism in scleroderma [77,83]. In SSc, the focus is traditionally on blood vessel dysfunction. However, affected SSc skin also shows lymphatic abnormalities with reduced micro-lymphatic networks associated with fingertip ulcers and disease progression [84,85,86]. Studies suggest that dermal lymphatic microangiopathy occurs due to blood capillary leaking leading to accumulation of fluids, inflammation, and fibrotic processes, as well as exacerbating micro-lymphatic damage [86]. This contributes to dystrophic changes and impaired tissue homeostasis in SSc. Ahmadzadeh, et al. [87] previously showed the effectiveness of the ADSC secretome for lymphangiogenesis in vitro. Other studies on fibrotic skin disease models found that ADSCs decrease the expression of adhesion molecules in ECs, including ICAM-1 and VCAM-1 [88], which facilitate the attachment of leukocytes and lead to their migration across the endothelium [89]. Both of these molecules are elevated in scleroderma [90]; however, the top interactions in this study cannot explain this effect (Figure 7). ADSCs may promote angiogenesis and especially lymphangiogenesis via VEGFD, MMP2, and Netrin 1, leading to revascularisation and normalisation of the disturbed microvascular architecture found in scleroderma.

We created a tractable model to assess how factors from ADSCs could interact with subtypes of fibroblasts and endothelial cells in scleroderma on the single-cell level. Understanding these interactions could generate a cell-free therapy by creating drugs to replicate these interactions. The model is especially useful because not only does research show a change in the ADSC phenotype in vitro, but also because in skin fibroblast cultures, certain subtypes, specifically the ECM-interacting fibroblasts, were observed to outgrow others [7,72]. Therefore, our in silico approach more closely mimics the interactions of ADSCs with all fibroblast populations in vivo. However, in our model, we estimate the effects of the secretome of ADSCs at the baseline level, unlike in vivo, where the secretory behaviour may be altered according to stimuli from the surrounding tissue and cells. It remains to be elucidated whether this alteration is essential for the anti-fibrotic and anti-inflammatory effects observed in vivo and in co-culture models in vitro. Further, it must be acknowledged that this analysis was conducted using two separate scRNA-seq datasets. While the dataset of SVF pertains to patients without systemic disease, the skin dataset comprises control and scleroderma patients. Unlike in a clinical scenario with autologous fat grafting, there is a disparity regarding age, gender, and the presence of SSc when simulating the interactions. This might affect the identified ligand/receptor interactions. Hence, these findings should not be extrapolated beyond their current scope. Apart from the direct effects on fibroblasts and ECs, the secretome of ADSCs possesses antioxidative, and therefore general anti-inflammatory capacity as well as the ability to restore the MMP/TIMP balance, which is deeply disturbed in scleroderma [91,92]. These effects, however, cannot be depicted in the interactome analysis. This study further highlights the importance of the vascular and perivascular niche for the pathophysiology of scleroderma. Vascular dysfunction with increased EC apoptosis and transendothelial migration of inflammatory cells may initiate a two-step process of fibroblast activation as proposed by Korsunsky, et al. [9]. ADSC secretome treatment may be able to disrupt vascular and perivascular inflammation hubs by altering the ECM composition of the perivascular niche and promoting effective angiogenesis through proangiogenic growth factors and decreasing EC apoptosis. The normalisation of EC behaviour would decrease chemotaxis and immune cell migration into the skin, while increased lymphangiogenesis may lead to more effective removal of immune cells. Effective lymphangiogenesis is heavily impaired in scleroderma [93,94]. However, this hypothesis needs further validation. Most interactions between ADSCs and both fibroblasts and ECs were based on proteins that are part of the ECM. The few effector molecules that were identified, including NTN1, VEGFD, MMP2, FGF2, and FNDC5, provide evidence for the anti-fibrotic and anti-inflammatory effects of the ADSC secretome (Figure 8); however, these few factors cannot explain all observed effects on scleroderma skin after autologous fat grafting. Additional studies on the single-cell level are needed to explore the interactions of ADSCs with other cell types in scleroderma skin, such as immune cells and keratinocytes—a pathway by which ADSCs could also indirectly affect fibroblast and EC behaviour. Further, we need to investigate the interactions of other fat graft-inherent cells on fibrotic skin to better understand the underlying mechanisms of autologous fat grafting as a treatment for scleroderma.

## 5. Conclusions

In conclusion, this study identified pro-inflammatory and ECM-interacting fibroblasts in SSc, which were also present in other chronic inflammatory diseases, suggesting a shared activation pathway. Various types of ECs presented a pro-fibrotic and pro-inflammatory transcriptome. The interactions between ADSCs and these cell types were primarily based on ECM proteins, and differential interactions with a potential therapeutic effect included NTN1, VEGFD, MMP2, FGF2, and FNDC5. The ADSC secretome may have the potential to disrupt perivascular inflammation hubs and promote angiogenesis and lymphangiogenesis in scleroderma. Key clinically beneficial phenomena observed after fat grafting may act through the ADSC secretome but remain unexplained. Though the elucidation of the effector mechanism or mechanisms has significant clinical potential in this and other fibrotic conditions.

## Figures and Tables

**Figure 1 cells-12-01784-f001:**
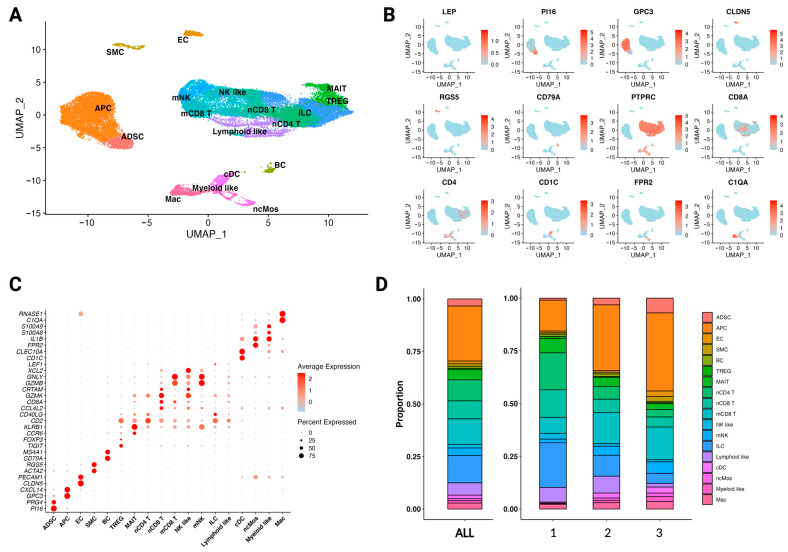
The SVF consists of various cell types with distinct transcriptomic signatures. (**A**) UMAP plot of 38,391 cells from the SVF of three healthy, non-obese patients (*n* = 3); 18 distinct clusters were obtained: adipose-derived stem cells (ADSC), adipocyte precursor cells (APC), endothelial cells (EC), smooth muscle cells (SMC), B cells (BC), regulatory T cells (TREG), mucosal-associated invariant T cells (MAIT), naïve CD4 T cells (nCD4 T), naïve CD8 T cells (nCD8 T), mature CD8 T cells (mCD8 T), natural killer-like cells (NK-like), mature natural killer cells (mNK), innate lymphoid cells (ILC), lymphoid-like cells (lymphoid-like), classical dendritic cells (cDC), non-classic monocytes (ncMos), myeloid-like cells (myeloid-like), and macrophages (Mac). (**B**) Feature plot of marker genes for cell type identification. Leptin (*LEP*), peptidase inhibitor 16 (*PI16*), glypican 3 (*GPC3*), claudin-5 (*CLDN5*), regulator of G-protein signalling 5 (*RGS5*), cluster of differentiation CD79A (*CD79A*), protein tyrosine phosphatase receptor type C (*PTPRC*), cluster of differentiation CD8A (*CD8A*), cluster of differentiation CD4 (*CD4*), cluster of differentiation CD1C (*CD1C*), formyl peptide receptor 2 (*FPR2*), and complement C1q A chain (*C1QA*). Expression of the gene is plotted onto the UMAP plot. The level of gene expression is indicated by colour intensity. (**C**) Dot plot of differentially expressed marker genes for each cluster, supporting cell type annotations shown in Figure 1A. Colour intensity implies the level of expression and dot size indicates the percentage of the cluster expressing the gene. (**D**) Stacked bar charts showing the proportion of each cell type in the SVF combined (left) and individually (right) for each patient.

**Figure 2 cells-12-01784-f002:**
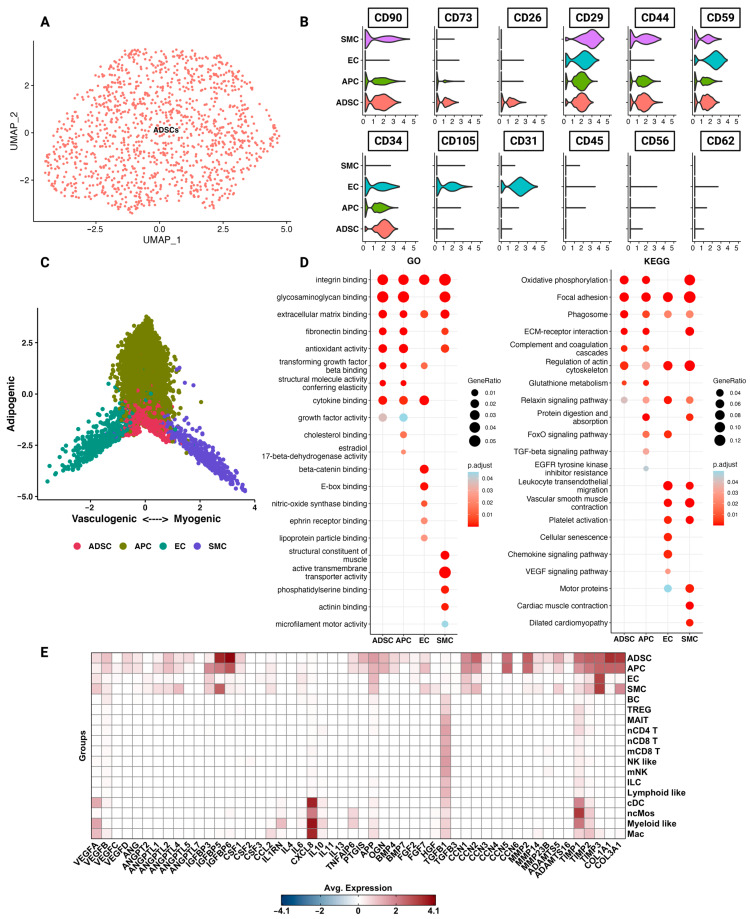
Adipose-derived stem cells (ADSCs) differ functionally and phenotypically from the rest of the SVF. (**A**) UMAP plot of 1285 cells identified as ADSCs from the SVF of three healthy, non-obese patients (*n* = 3). (**B**) Violin plots of structural cells showing the cluster expression of commonly used cell surface markers used to characterise ADSCs. (**C**) Cell state plot based on enrichment scores for gene combinations indicating commitment to adipogenic (*APOD*, *IGF1*, and *CXCL14*), vasculogenic (*ENG*, *PECAM1*), and myogenic (*ACTA2*, *RGS5*, and *TAGLN*) lineage. The more negative or more positive the score, the higher the enrichment for genes associated with the distinct cell lineages. (**D**) Results of pathway enrichment analysis for gene ontology (GO) and the Kyoto Encyclopaedia of Genes and Genomes (KEGG) shown as dot plots. Red colour indicates higher significance. Larger dot size indicates a higher gene ratio. (**E**) Heatmap of additional genes of interest identified through differential gene expression analysis (DEG) and literature review. Red colour = high expression, and blue colour = low expression.

**Figure 3 cells-12-01784-f003:**
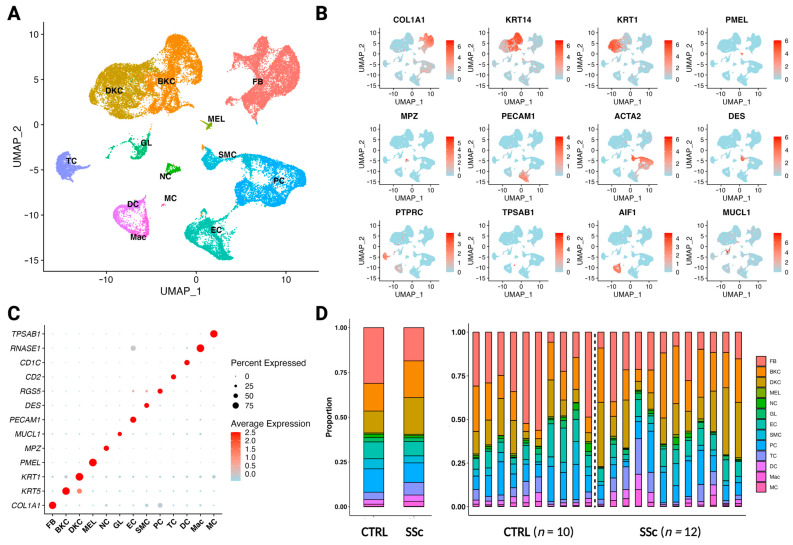
Differences in single-cell composition between healthy and scleroderma skin. (**A**) UMAP plot of 37,958 cells from skin biopsies of ten control patients (*n* = 10) and twelve scleroderma patients with diffuse cutaneous disease (*n* = 12). A total of 13 clusters were obtained: fibroblasts (FB), basal keratinocytes (BKC), differentiating keratinocytes (DKC), endothelial cells (EC), smooth muscle cells (SMC), T cells (TC), dendritic cells (DC), macrophages (Mac), and mast cells (MC). (**B**) Feature plot of marker genes for cell type identification. Collagen type I alpha 1 chain (*COL1A1*), keratin 14 (*KRT14*), keratin 1 (*KRT1*), premelanosome protein (*PMEL*), myelin protein zero (*MPZ*), platelet and endothelial cell adhesion molecule 1 (*PECAM1*), smooth muscle actin alpha 2 (*ACTA2*), desmin (*DES*), tyrosine phosphatase receptor type C (*PTPRC*), tryptase alpha/beta 1 protein (*TPSAB1*), allograft inflammatory factor 1 (*AIF1*), and mucin-like 1 protein (*MUCL1*). Expression of the gene is plotted onto the UMAP plot. The level of gene expression is indicated by colour intensity. (**C**) Dot plot of differentially expressed marker genes for each cluster, supporting cell type annotations shown in Figure 3A. Colour intensity implies level of expression and dot size indicates the percentage of the cluster expressing the gene. (**D**) Stacked bar charts showing the proportion of each cell type in control versus scleroderma skin combined (left) and individually (right) for each patient.

**Figure 4 cells-12-01784-f004:**
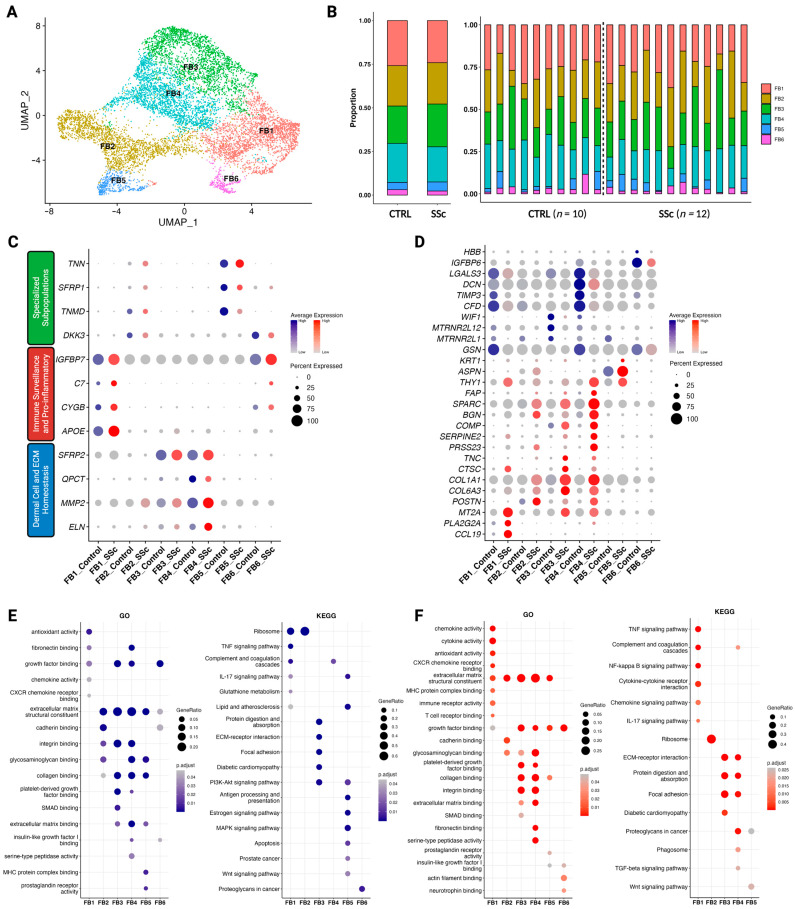
Dermal fibroblast subclusters exhibit distinct functionalities in control versus scleroderma. (**A**) UMAP plot of 9192 cells identified as fibroblasts from skin biopsies of ten control patients (*n* = 10) and twelve scleroderma patients with diffuse cutaneous disease (*n* = 12). Six subclusters were obtained (FB1-FB6). (**B**) Stacked bar charts showing the proportion of each cell type in control versus scleroderma skin combined (left) and individually (right) for each patient. (**C**) Dot plot of marker genes for each cluster according to the classification system by Ascensión, et al. [7]. Colour intensity implies level of expression and dot size indicates the percentage of the cluster expressing the gene (blue = control; red = scleroderma). (**D**) Dot plot of differentially expressed marker genes for each cluster when comparing between subclusters and between control and scleroderma. Colour intensity implies level of expression and dot size indicates the percentage of the cluster expressing the gene (blue = control; red = scleroderma). (**E**,**F**) Results of pathway enrichment analysis for gene ontology (GO) and the Kyoto Encyclopaedia of Genes and Genomes (KEGG) shown as dot plots for control (**E**) and scleroderma (**F**). Colour intensity indicates higher significance. Larger dot size indicates a higher gene ratio. For FB6_SSc_, no KEGG pathways were enriched (**F**).

**Figure 5 cells-12-01784-f005:**
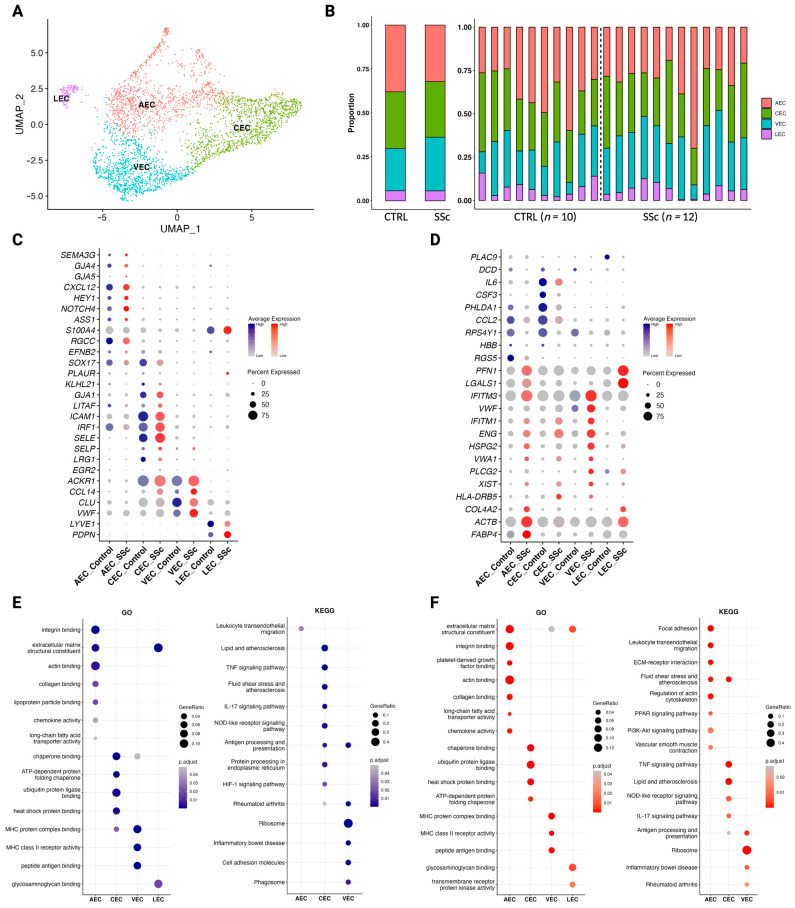
Dermal endothelial cell subclusters exhibit distinct functionalities in control versus scleroderma. (**A**) UMAP plot of 3282 cells identified as endothelial cells from skin biopsies of ten control patients (*n* = 10) and twelve scleroderma patients with diffuse cutaneous disease (*n* = 12). Four subclusters were obtained: arterial endothelial cells (AEC), capillary endothelial cells (CEC), venous endothelial cells (VEC), and lymphatic endothelial cells (LEC). (**B**) Stacked bar charts showing the proportion of each cell type in control versus scleroderma skin combined (left) and individually (right) for each patient. (**C**) Dot plot of marker genes for each cluster according to the classification system by He, et al. [37]. Colour intensity implies level of expression and dot size indicates the percentage of the cluster expressing the gene (blue = control; red = scleroderma). (**D**) Dot plot of differentially expressed marker genes for each cluster when comparing between subclusters and between control and scleroderma. Colour intensity implies level of expression and dot size indicates the percentage of the cluster expressing the gene (blue = control; red = scleroderma). (**E**,**F**) Results of pathway enrichment analysis for gene ontology (GO) and the Kyoto Encyclopaedia of Genes and Genomes (KEGG) shown as dot plots for control (**E**) and scleroderma (**F**). Colour intensity indicates higher significance. Larger dot size indicates a higher gene ratio. For LEC, no KEGG pathways were enriched in control (**E**) and scleroderma (**F**).

**Figure 6 cells-12-01784-f006:**
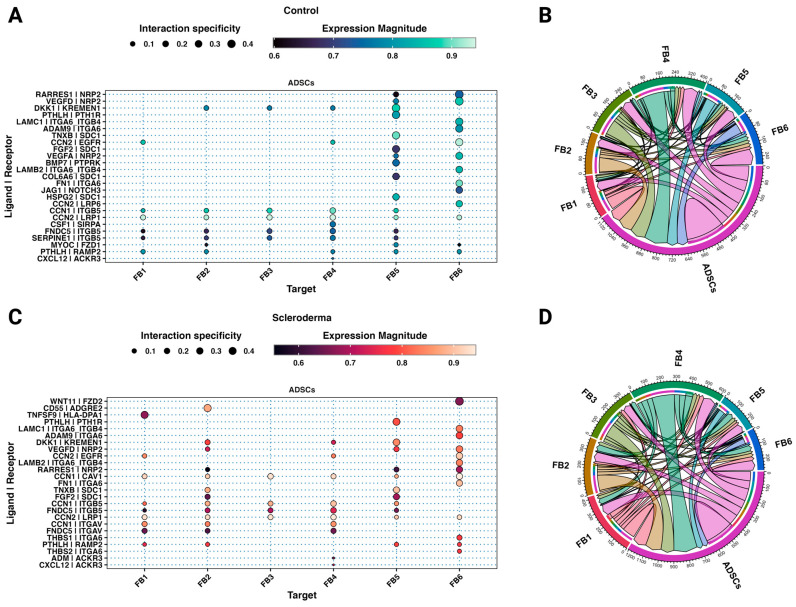
ADSCs interact distinctly with fibroblasts from scleroderma. (**A**,**C**) Dot plots showing the top 25 unidirectional cell–cell ligand–receptor interactions between ADSCs (ligands) and fibroblast subtypes (receptors) from control (**A**) and scleroderma patients (**C**) based on aggregate ranks. Colour implies expression magnitude and dot size indicates the specificity of interaction. (**B**,**D**) Chord diagrams showing the top 50 bilateral cell–cell ligand–receptor interactions between all cell types from control (**B**) and scleroderma patients (**D**) based on aggregate ranks.

**Figure 7 cells-12-01784-f007:**
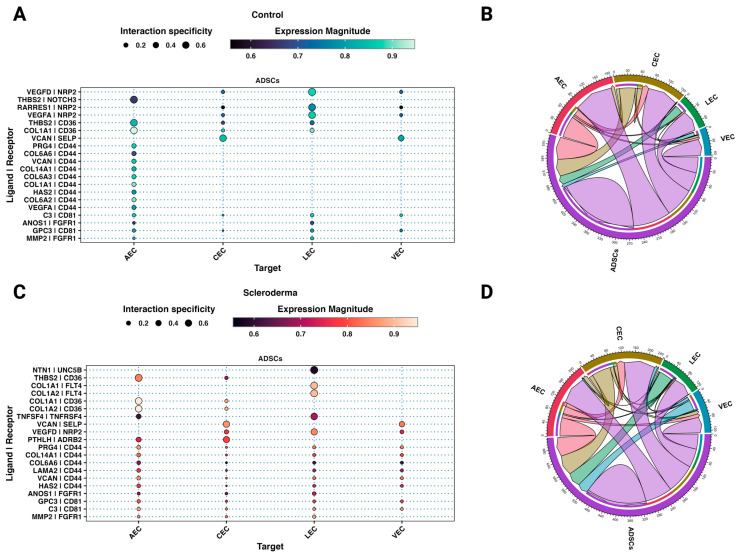
ADSCs interact distinctly with endothelial cells from scleroderma. (**A**,**C**) Dot plots showing the top 25 unidirectional cell–cell ligand–receptor interactions between ADSCs (ligands) and fibroblast subtypes (receptors) from control (**A**) and scleroderma patients (**C**) based on aggregate ranks. Colour implies expression magnitude and dot size indicates the specificity of interaction. (**B**,**D**) Chord diagrams showing the top 50 bilateral cell–cell ligand–receptor interactions between all cell types from control (**B**) and scleroderma patients (**D**) based on aggregate ranks.

**Figure 8 cells-12-01784-f008:**
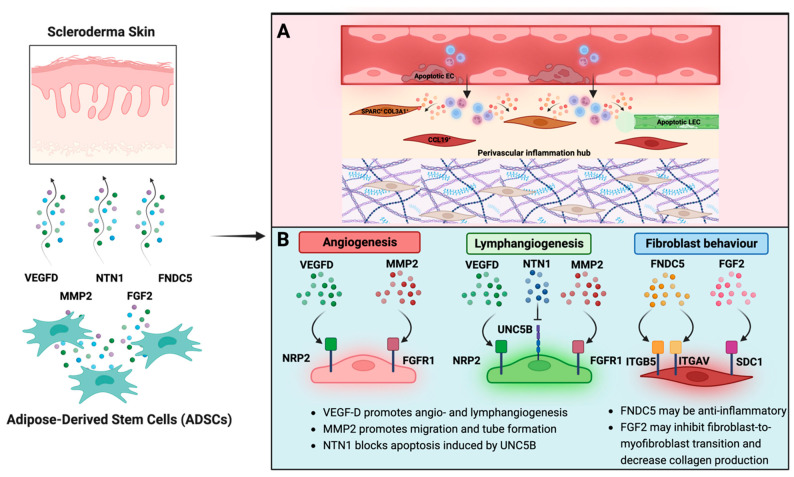
Summary of possibly therapeutic interactions of ADSCs with fibroblasts and endothelial cells from scleroderma. (**A**) Single-cell analyses of chronic inflammatory diseases [9] and scleroderma [8] identified a proinflammatory CCL19^+^ and a SPARC^+^/COL3A1^+^ fibroblast subset that colocalises with the vasculature, highlighting the importance of the perivascular inflammation hub for activation of fibroblasts in scleroderma. (**B**) Based on differences and commonalities in cell-to-cell receptor–ligand interactions of ADSCs with fibroblasts and endothelial cells from healthy and scleroderma skin, *VEGFD, MMP2, NTN1, FNDC5,* and *FGF2* were identified as possible anti-fibrotic effector molecules. Created with BioRender.com.

## Data Availability

The scRNA-seq datasets analysed during the current study are available from the Gene Expression Omnibus (GEO) with accession nos. GSE155960 [25] and GSE138669 [8]. Code is available upon reasonable request. This study did not generate new unique reagents.

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
