# Peer review of "Single-Cell Analysis of ADSC Interactions with Fibroblasts and Endothelial Cells in Scleroderma Skin"

_cells, 2023, doi:10.3390/cells12131784_

Round 1

Reviewer 1 Report

Nice work. 

I have a few things that could benefit the reader. 

1) Is it possible to specify the dcSSc patients autoantibody background

2) Is it possible to indicate the degree of severity of the dcSSc patients eg Rodnan skin score and if the biopsies are taken in a fibrotic area.

3) It puzzles me a bit that the authors write that endothelia cells do not express high levels of CD73 (Fig 2B).  Please comment.

Reviewer 2 Report

This is an interesting paper using in silico analysis to gain insight on the possible mechanisms involved in the beneficial effect of ADSCs in scleroderma. 

(1) For figure 2, figure 2C was not mentioned. The ADSCs does not seem to separate out from the 3 other categories. How do the authors explain this?

(2) In the paragraph of describing Figure 2 (under the transcriptomic profile and functions of adipose-derived stem cells), there are some mislabeling going on: the second Figure 2A mentioned should be Figure 2B. Similarly, the second Figure 2D mentioned should be Figure 2E.

(3) For Figure 2B, it is interesting that the ADSC marker CD105 has minimal abundance while CD34, which is not a ACSC marker, was high. The authors speculated the reason for this in the discussion, but it will be more clear if the authors mention that here in the results, or discuss this discrepancy specifically in the discussion. Other ADSC markers such as CD59, CD44, and CD29, and non ADSC markers such as CD56, CD62, or HLA molecules can also be included.

(4) It would be nice to show the GO and KEGG pathways also for APC, EC, and SMC in Figure 2D.

(5) The discussion is rather long and hard to follow with so many pathways and genes mentioned. Suggest to shorten it and focus on the most important points.

(6) The authors should state the limitation of this in silico approach. This type of analysis is nice, but it is mainly descriptive. The interaction analysis is the most critical part of this paper, but this analysis is done using two different single-cell RNA-seq datasets from healthy controls vs. controls/scleroderma with different ages between the two. Whether these ligand/receptor pairs at the transcript level identified realistically reflect disease pathways is questionable. The readers should be warned to be cautious of extrapolating these results.

Reviewer 3 Report

In this manuscript, the authors reprocessed and analyzed the single-cell RNA Sequencing data from the GEO and then simulated the interaction of ADSCs with fibroblasts and endothelial cells from scleroderma skin in silico. They found that the interaction between ADSCs and these cell types were primarily based on ECM proteins, and that the ADSC secretome may disrupt vascular and perivascular inflammation hubs by promoting angiogenesis and lymphangiogenesis in scleroderma. This work sounds interesting and meaningful. There are a few comments.

1. Page 3, line 20, authors would be better off using “healthy controls subjects” (the expression of controls in the original data source) rather than “ten patients”, as it may easily mislead the readers.

2. Page 6, line 20-21, this sentence is a description of Figure 2B, not Figure 2A.

3. Page 6, line 36-38, this sentence is a description of Figure 2E, not Figure 2D, and ANGPTL7 was not enriched in ADSCs in Figure 2E.

4. Page 8, line 8-10, the cells from skin biopsies of ten controls and twelve scleroderma patients were clustered into 13 clusters, but only nine clusters were described. Please provide the expanded form of the other four clusters (PC, NC, GL, and MEL).

5. Page 13, line 19, authors should provide the expanded form of ITGAV and CAV1 when they first appeared.

6. Page 16, line 10, “Sup Figure 7” was written incorrectly.

7. In Figure 4A, it showed six clusters, but the authors wrote “five clusters” in the legend. Please describe it accurately.

8. The KEGG analysis results of FB6 (in Figure 4F) and LEC (in Figure 5E-F) were missing.

9. The legends in the supplementary were not consistent with the corresponding positions marked in the manuscript, such as in Figure S1: These DEGs support the clustering seen in Fig 1, not Fig 3. Please check carefully to maintain consistency.
